# TEXT-TO-3D WITH CLASSIFIER SCORE DISTILLATION

**Xin Yu**[1]* **Yuan-Chen Guo**[2,3] **Yangguang Li**[3] **Ding Liang**[3] **Song-Hai Zhang**[2] **Xiaojuan Qi**[1]†
[1]The University of Hong Kong    [2]Tsinghua University    [3]VAST

## ABSTRACT

Text-to-3D generation has made remarkable progress recently, particularly with methods based on Score Distillation Sampling (SDS) that leverages pre-trained 2D diffusion models. While the usage of classifier-free guidance is well acknowledged to be crucial for successful optimization, it is considered an auxiliary trick rather than the most essential component. In this paper, we re-evaluate the role of classifier-free guidance in score distillation and discover a surprising finding: the guidance alone is enough for effective text-to-3D generation tasks. We name this method *Classifier Score Distillation (CSD)*, which can be interpreted as using an implicit classification model for generation. This new perspective reveals new insights for understanding existing techniques. We validate the effectiveness of CSD across a variety of text-to-3D tasks including shape generation, texture synthesis, and shape editing, achieving results superior to those of state-of-the-art methods. Our project page is https://xinyu-andy.github.io/Classifier-Score-Distillation

## 1 INTRODUCTION

3D content creation is important for many applications, such as interactive gaming, cinematic arts, AR/VR, and simulation. However, it is still challenging and expensive to create a high-quality 3D asset as it requires a high level of expertise. Therefore, automating this process with generative models has become an important problem, which remains challenging due to the scarcity of data and the complexity of 3D representations.

Recently, techniques based on Score Distillation Sampling (SDS) (Poole et al., 2022; Lin et al., 2023; Chen et al., 2023; Wang et al., 2023b), also known as Score Jacobian Chaining (SJC) (Wang et al., 2023a), have emerged as a major research direction for text-to-3D generation, as they can produce high-quality and intricate 3D results from diverse text prompts without requiring 3D data for training. The core principle behind SDS is to optimize 3D representations by encouraging their rendered images to move towards high probability density regions conditioned on the text, where the supervision is provided by a pre-trained 2D diffusion model (Ho et al., 2020; Sohl-Dickstein et al., 2015; Rombach et al., 2022; Saharia et al., 2022; Balaji et al., 2022). DreamFusion (Poole et al., 2022) advocates the use of SDS for the optimization of Neural Radiance Fields (NeRF). (Barron et al., 2022; Mildenhall et al., 2021). Subsequent research improve the visual quality by introducing coarse-to-fine optimization strategies (Lin et al., 2023; Chen et al., 2023; Wang et al., 2023b), efficient 3D representations (Lin et al., 2023; Chen et al., 2023), multi-view-consistent diffusion models (Shi et al., 2023; Zhao et al., 2023), or new perspectives on modeling the 3D distribution (Wang et al., 2023b). Despite these advancements, all these works fundamentally rely on score distillation for optimization.

Although score distillation is theoretically designed to optimize 3D representations through probability density distillation (Poole et al., 2022; Oord et al., 2018), as guided by pre-trained 2D diffusion models (i.e., Eq. (4)), a practical gap emerges in its implementation due to the widespread reliance on classifier-free guidance (CFG) (Ho & Salimans, 2022). When using CFG, the gradient that drives the optimization actually comprises two terms. The primary one is the gradient of the log data density, i.e., $\log p_\phi(\mathbf{x}_t|y)$, estimated by the diffusion models to help move the synthesized images $\mathbf{x}$ to high-density data regions conditioned on a text prompt $y$, which is the original optimization objective (see Eqs. (4) and (5)). The second term is the gradient of the log function of the posterior,

---

*This work is in collaboration with VAST.
†Corresponding author.

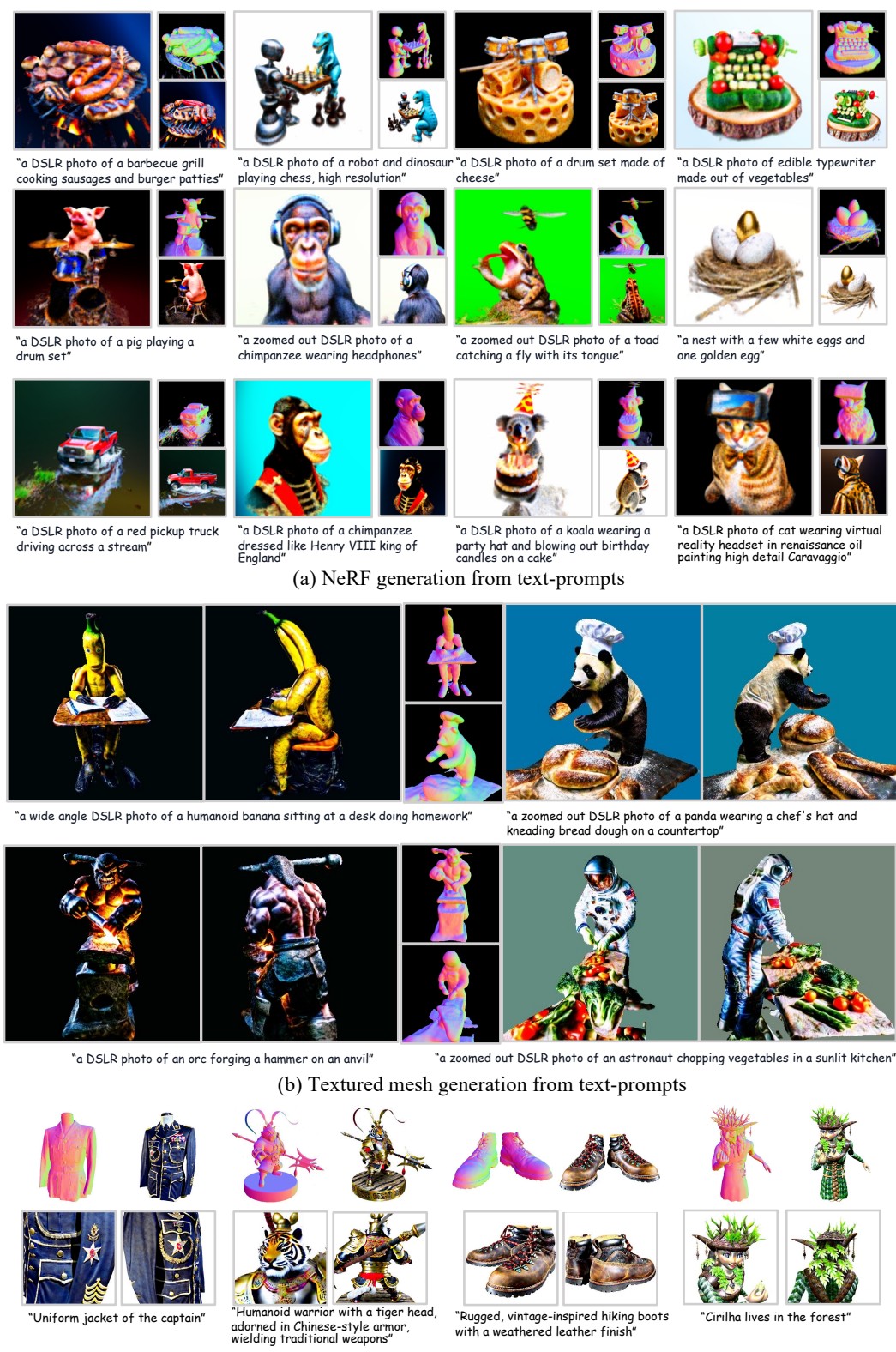

(a) NeRF generation from text-prompts

(b) Textured mesh generation from text-prompts

(c) Synthesize texture for given meshes from text-prompts

Figure 1: Illustrative overview of our method's capabilities. (a) Generation of a Neural Radiance Field (NeRF) from text inputs, trained using a 64×64 diffusion guidance model. (b) Subsequent refinement leads to high-quality textured meshes using a 512×512 diffusion guidance model. (c) Texture synthesis on user-specified meshes, resulting in highly realistic and high-resolution detail.

i.e., $\log p_\phi(y|\mathbf{x}_t)$, which can be empirically interpreted as an implicit classification model (Ho & Salimans, 2022). We elaborate more on this in Sec. 3. The combination of these two components determines the final effect of optimizing the 3D representation. Notably, although the use of CFG (i.e., the second term) is widely recognized as vital for text-to-3D generation, it is considered an auxiliary trick to help the optimization, rather than as the primary driving force.

In this paper, we show that the classifier component of Score Distillation Sampling (SDS) is not just auxiliary but essential for text-to-3D generation, and using only this component is sufficient. We call this paradigm *Classifier Score Distillation (CSD). This insight fundamentally shifts our understanding of the mechanisms underlying the success of text-to-3D generation based on score distillation. Specifically, the efficacy stems from distilling knowledge from an implicit classifier $p_\phi(y|\mathbf{x}_t)$ rather than from reliance on the generative prior $p_\phi(\mathbf{x}_t|y)$.* This finding further enables us to uncover fresh insights into existing design choices: 1) We demonstrate that utilizing negative prompts can be viewed as a form of joint optimization with dual classifier scores. This realization enables us to formulate an annealed negative classifier score optimization strategy, which enhances generation quality while maintaining the result faithfulness according to prompts. 2) Building on our understanding, we illustrate the utilization of classifier scores for efficient text-driven 3D editing. 3) The Variational Score Distillation technique (Wang et al., 2023b) can be viewed as an adaptive form of negative classifier score optimization, where the negative direction is supplied by a diffusion model trained concurrently.

CSD can be seamlessly integrated into existing SDS-based 3D generation pipelines and applications, such as text-driven NeRF generation, textured mesh generation, and texture synthesis. As demonstrated in Fig. 1, our method produces high-quality generation results, featuring extremely photo-realistic appearances and the capability to generate scenes corresponding to complex textual descriptions. We conduct extensive experiments to evaluate the robustness of our approach and compare our method against existing methods, achieving state-of-the-art results.

## 2 DIFFUSION MODELS

The diffusion model (Sohl-Dickstein et al., 2015; Ho et al., 2020; Song et al., 2021; Song & Ermon, 2020) is a type of likelihood-based generative model used for learning data distributions. Given an underlying data distribution $q(\mathbf{x})$, the model initiates a forward process that progressively adds noise to the data $\mathbf{x}$ sampled from $q(\mathbf{x})$. This produces a sequence of latent variables $\{\mathbf{x}_0 = \mathbf{x}, \mathbf{x}_1, \dots, \mathbf{x}_T\}$. The process is defined as a Markov Chain $q(\mathbf{x}_{1:T}|\mathbf{x}) := \prod_{t=1}^{T} q(\mathbf{x}_t|\mathbf{x}_{t-1})$ with Gaussian transition kernels. The marginal distribution of latent variables at time $t$ is given by $q(\mathbf{x}_t|\mathbf{x}) = \mathcal{N}(\alpha_t\mathbf{x}, \sigma_t^2\mathbf{I})$. This is equivalent to generating a noise sample $\mathbf{x}_t$ using the equation $\mathbf{x}_t = \alpha_t\mathbf{x} + \sigma_t\epsilon$, where $\epsilon \sim \mathcal{N}(\mathbf{0}, \mathbf{I})$. Parameters $\alpha_t$ and $\sigma_t$ are chosen such that $\sigma_t^2 + \alpha_t^2 = 1$, and $\sigma_t$ gradually increases from 0 to 1. Thus, $q(\mathbf{x}_t)$ converges to a Gaussian prior distribution $\mathcal{N}(\mathbf{0}, \mathbf{I})$.

Next, a reverse process (i.e., the generative process) is executed to reconstruct the original signal from $\mathbf{x}_T$. This is described by a Markov process $p_\phi(\mathbf{x}_{0:T}) := p(\mathbf{x}_T)\prod_{t=1}^{T} p_\phi(\mathbf{x}_{t-1}|\mathbf{x}_t)$, with the transition kernel $p_\phi(\mathbf{x}_{t-1}|\mathbf{x}_t) := \mathcal{N}(\mu_\phi(\mathbf{x}_t, t), \sigma_t^2\mathbf{I})$. The training objective is to optimize $\mu_\phi$ by maximizing a variational lower bound of the log data likelihood. In practice, $\mu_\phi$ is re-parameterized as a denoising network $\epsilon_\phi$ (Ho et al., 2020) and the loss can be further simplified to a Mean Squared Error (MSE) criterion (Ho et al., 2020; Kingma et al., 2021):

$$\mathcal{L}_{\text{Diff}}(\phi) := \mathbb{E}_{\mathbf{x}\sim q(\mathbf{x}), t\sim\mathcal{U}(0,1), \epsilon\sim\mathcal{N}(\mathbf{0},\mathbf{I})}\left[\omega(t)\|\epsilon_\phi(\alpha_t\mathbf{x} + \sigma_t\epsilon; t) - \epsilon\|_2^2\right], \quad (1)$$

where $w(t)$ is time-dependent weights.

The objective function can be interpreted as predicting the noise $\epsilon$ that corrupts the data $\mathbf{x}$. Besides, it is correlated to NCSN denoising score matching model (Ho et al., 2020; Song & Ermon, 2020), and thus the predicted noise is also related to the score function of the perturbed data distribution $q(\mathbf{x}_t)$, which is defined as the gradient of the log-density with respect to the data point:

$$\nabla_{\mathbf{x}_t}\log q(\mathbf{x}_t) \approx -\epsilon_\phi(\mathbf{x}_t; t)/\sigma_t. \quad (2)$$

This means that the diffusion model can estimate a direction that guides $\mathbf{x}_t$ towards a high-density region of $q(\mathbf{x}_t)$, which is the key idea of SDS for optimizing the 3D scene. Since samples in high-

density regions of $q(\mathbf{x}_t)$ are assumed to reside in the high-density regions of $q(\mathbf{x}_{t-1})$ (Song & Ermon, 2020), repeating the process can finally obtain samples of good quality from $q(\mathbf{x})$.

**Classifier-Free Guidance**  Text-conditioned diffusion models (Balaji et al., 2022; Saharia et al., 2022; Rombach et al., 2022; Ramesh et al., 2022) generate samples $\mathbf{x}$ based on a text prompt $y$, which is also fed into the network as an input, denoted as $\epsilon_\phi(\mathbf{x}_t; y, t)$. A common technique to improve the quality of these models is classifier-free guidance (CFG) (Ho & Salimans, 2022). This method trains the diffusion model in both conditioned and unconditioned modes, enabling it to estimate both $\nabla_{\mathbf{x}_t} \log q(\mathbf{x}_t|y)$ and $\nabla_{\mathbf{x}_t} \log q(\mathbf{x}_t)$, where the network are denoted as $\epsilon_\phi(\mathbf{x}_t; y, t)$ and $\epsilon_\phi(\mathbf{x}_t; t)$. During sampling, the original score is modified by adding a guidance term, i.e., $\epsilon_\phi(\mathbf{x}_t; y, t) \rightarrow \epsilon_\phi(\mathbf{x}_t; y, t) + \omega \cdot [\epsilon_\phi(\mathbf{x}_t; y, t) - \epsilon_\phi(\mathbf{x}_t; t)]$, where $\omega$ is the guidance scale that controls the trade-off between fidelity and diversity. Using Eq. (2) and Bayes' rule, we can derive the following relation:

$$-\frac{1}{\sigma_t}[\epsilon_\phi(\mathbf{x}_t; y, t) - \epsilon_\phi(\mathbf{x}_t; t)] = \nabla_{\mathbf{x}_t} \log q(\mathbf{x}_t|y) - \nabla_{\mathbf{x}_t} \log q(\mathbf{x}_t) \propto \nabla_{\mathbf{x}_t} \log q(y|\mathbf{x}_t). \quad (3)$$

Thus, the guidance can be interpreted as the gradient of an implicit classifier (Ho & Salimans, 2022).

## 3  WHAT MAKES SDS WORK?

Score Distillation Sampling (SDS) (Poole et al., 2022) is a novel technique that leverages pretrained 2D diffusion models for text-to-3D generation. SDS introduces a loss function, denoted as $\mathcal{L}_{\text{SDS}}$, whose gradient is defined as follows:

$$\nabla_\theta \mathcal{L}_{\text{SDS}}(g(\theta)) = \mathbb{E}_{t,\epsilon,\mathbf{c}}\left[w(t)\frac{\sigma_t}{\alpha_t}\nabla_\theta \text{KL}\left(q(\mathbf{x}_t|\mathbf{x} = g(\theta; \mathbf{c})) \| p_\phi(\mathbf{x}_t|y)\right)\right], \quad (4)$$

where $\theta$ is the learnable parameter of a 3D representation (e.g., NeRF), and $g$ is a differentiable rendering function that enables obtaining the rendered image from the 3D scene $\theta$ and camera $\mathbf{c}$. The optimization aims to find modes of the score functions that are present across all noise levels throughout the diffusion process which is inspired by work on probability density distillation (Poole et al., 2022; Oord et al., 2018).

Eq. (4) can be simplified into: $\mathbb{E}_{t,\epsilon,\mathbf{c}}\left[w(t)(\epsilon_\phi(\mathbf{x}_t; y, t))\frac{\partial \mathbf{x}}{\partial \theta}\right]$. This formulation is more intuitive than Eq. (4): The conditional score function $\epsilon_\phi(\mathbf{x}_t; y, t)$ indicates the gradient for updating $\mathbf{x}_t$ to be closer to high data density regions (see Eq. (2)), allowing the parameter $\theta$ to be trained using the chain rule. However, in practice, DreamFusion recommends adjusting the gradient $\epsilon_\phi(\mathbf{x}_t; y, t)$ to $\epsilon_\phi(\mathbf{x}_t; y, t) - \epsilon$ since this modification has been found to improve convergence. This change does not alter the overall objective because $\mathbb{E}_{t,\epsilon,\mathbf{c}}\left[w(t)(-\epsilon)\frac{\partial \mathbf{x}}{\partial \theta}\right] = 0$. Consequently, the final gradient to update $\theta$ is expressed as:

$$\nabla_\theta \mathcal{L}_{\text{SDS}} = \mathbb{E}_{t,\epsilon,\mathbf{c}}\left[w(t)(\epsilon_\phi(\mathbf{x}_t; y, t) - \epsilon)\frac{\partial \mathbf{x}}{\partial \theta}\right]. \quad (5)$$

**What Makes SDS Work?**  A crucial aspect of score distillation involves computing the gradient to be applied to the rendered image $\mathbf{x}$ during optimization, which we denote as $\delta_x(\mathbf{x}_t; y, t) := \epsilon_\phi(\mathbf{x}_t; y, t) - \epsilon$. This encourages the rendered images to reside in high-density areas conditioned on the text prompt $y$. From Eq. (4), in principle, $\epsilon_\phi(\mathbf{x}_t; y, t)$ should represent the pure text-conditioned score function. However, in practice, classifier-free guidance is employed in diffusion models with a large guidance weight $\omega$ (e.g., $\omega = 100$ in DreamFusion (Poole et al., 2022)) to achieve high-quality results, causing the final gradient applied to the rendered image to deviate from Eq. (5). Specifically, $\delta_x$ with CFG is expressed as:

$$\delta_x(\mathbf{x}_t; y, t) = \underbrace{[\epsilon_\phi(\mathbf{x}_t; y, t) - \epsilon]}_{\delta_x^{\text{gen}}} + \omega \cdot \underbrace{[\epsilon_\phi(\mathbf{x}_t; y, t) - \epsilon_\phi(\mathbf{x}_t; t)]}_{\delta_x^{\text{cls}}}. \quad (6)$$

The gradient can be decomposed into two parts, i.e., $\delta_x := \delta_x^{\text{gen}} + \omega \cdot \delta_x^{\text{cls}}$. According to Eq. (2) and Eq. (3), the first term $\delta_x^{\text{gen}}$ is associated with $\nabla_{\mathbf{x}_t} \log q(\mathbf{x}_t|y)$. This term signifies the gradient direction in which the image should move to become more realistic conditioned on the text, which we refer to as the generative prior. $\delta_x^{\text{cls}}$ is related to $\nabla_{\mathbf{x}_t} \log q(y|\mathbf{x}_t)$, representing the update direction required for the image to align with the text evaluated by an implicit classifier model, which emerges from the diffusion model (Ho & Salimans, 2022). We refer to this part as the classifier score.

Thus, there is a gap between the theoretical formulation in Eq. (4) and the practical implementation in Eq. (6). To better understand the roles of different terms, we validate their contribution by visualizing the gradient norm throughout the training of a NeRF via SDS with guidance scale $\omega = 40$. As shown in Fig. 2 (a), the gradient norm of the generative prior is several times larger than that of the classifier score. However, to generate high-quality results, a large guidance weight must be set, as shown in Fig. 2 (b). When incorporating both components, the large guidance weight actually causes the gradient from the classifier score to dominate the optimization direction. Moreover, the optimization process fails when relying solely on the generative component, as indicated by setting $\omega = 0$ in Eq. (6). This observation leads us to question **whether the classifier score is the true essential component that drives the optimization**. To explore this, we introduce Classifier Score Distillation (CSD), which employs only the classifier score for optimization.

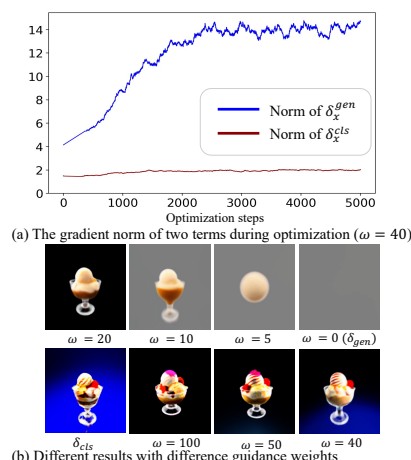

(a) The gradient norm of two terms during optimization ($\omega = 40$)

$\omega = 20$  $\omega = 10$  $\omega = 5$  $\omega = 0$ ($\delta_{gen}$)

$\delta_{cls}$  $\omega = 100$  $\omega = 50$  $\omega = 40$
(b) Different results with difference guidance weights

Figure 2: (a) The gradient norm during optimization. (b) Optimization results through different guidance weights.

## 4 CLASSIFIER SCORE DISTILLATION (CSD)

Consider a text prompt $y$ and a parameterized image generator $g(\theta)$. We introduce a loss function denoted as Classifier Score Distillation $\mathcal{L}_{\text{CSD}}$, the gradient of which is expressed as follows:

$$\nabla_\theta \mathcal{L}_{\text{CSD}} = \mathbb{E}_{t,\epsilon,\mathbf{c}} \left[ w(t)(\epsilon_\phi(\mathbf{x}_t; y, t) - \epsilon_\phi(\mathbf{x}_t; t)) \frac{\partial \mathbf{x}}{\partial \theta} \right]. \tag{7}$$

Our loss is fundamentally different from the previous principle presented in Eq. (4). According to Eq. (3), we use an implicit classification model that is derived from the generative diffusion models to update the 3D scene. Specifically, text-to-image diffusion models with CFG can generate both conditioned and unconditioned outputs, and we can derive a series of implicit noise-aware classifiers (Ho & Salimans, 2022) for different time steps $t$, based on Eq. (3). These classifiers can evaluate the alignment between the noisy image and the text $y$. Thus, the overall optimization objective in Eq. (7) seeks to refine the 3D scene in such a manner that rendered images at any noise level $t$ align closely with their respective noise-aware implicit classifiers. We will use $\delta_x^{\text{cls}}(\mathbf{x}_t; y, t) := \epsilon_\phi[(\mathbf{x}_t; y, t) - \epsilon_\phi(\mathbf{x}_t; t)]$ to denote the classifier score for further discussions.

We note that some previous works have also used text and image matching losses for 3D generation, rather than relying on a generative prior, such as the CLIP loss (Jain et al., 2022; Michel et al., 2022). However, these methods are still less effective than score distillation methods based on the generative diffusion model. In the next section, we show that our loss function can produce high-quality results that surpass those using the SDS loss. Moreover, our formulation enables more flexible optimization strategies and reveals new insights into existing techniques. In the following, we first introduce improved training strategies based on CSD, then we develop the application for 3D editing and discuss the connections between CSD and Variational Score Distillation (VSD).

**CSD with Annealed Negative Prompts** We observe that employing negative prompts $y_{\text{neg}}$, text queries describing undesired images, can accelerate the training process and enhance the quality of the generated results. We provide an interpretation from the perspective of CSD. Specifically, with negative prompts, the SDS loss can be rewritten as:

$$\delta_x^{\text{sds}} = [\epsilon_\phi(\mathbf{x}_t; y, t) - \epsilon] + \omega \cdot [\epsilon_\phi(\mathbf{x}_t; y, t) - \epsilon_\phi(\mathbf{x}_t; y_{\text{neg}}, t)] \tag{8}$$

Recall that $\delta_x^{\text{cls}}(\mathbf{x}_t; y, t) := \epsilon_\phi[(\mathbf{x}_t; y, t) - \epsilon_\phi(\mathbf{x}_t; t)]$, we arrive at the following expression:

$$\begin{aligned}
\epsilon_\phi(\mathbf{x}_t; y, t) - \epsilon_\phi(\mathbf{x}_t; y_{\text{neg}}, t) &= [\epsilon_\phi(\mathbf{x}_t; y, t) - \epsilon_\phi(\mathbf{x}_t; t)] - [\epsilon_\phi(\mathbf{x}_t; y_{\text{neg}}, t) - \epsilon_\phi(\mathbf{x}_t; t)] \\
&= \delta_x^{\text{cls}}(\mathbf{x}_t; y, t) - \delta_x^{\text{cls}}(\mathbf{x}_t; y_{\text{neg}}, t)
\end{aligned} \tag{9}$$

Therefore, the use of negative prompts can be seen as a dual-objective Classifier Score Distillation: It not only pushes the model toward the desired prompt but also pulls it away from the unwanted

states, evaluated by two classifier scores. However, this may also cause a trade-off between quality and fidelity, as the negative prompts may make the optimization diverge from the target text prompt $y$, resulting in mismatched text and 3D outputs, as shown in Fig. 5.

Given CSD and our novel insights into the effects of negative prompts, we introduce an extended CSD formulation that redefines $\delta_x^{\text{cls}}$ as follows:

$$
\begin{aligned}
\delta_x^{\text{cls}} =& \omega_1 \cdot \delta_x^{\text{cls}}(\mathbf{x}_t; y, t) - \omega_2 \cdot \delta_x^{\text{cls}}(\mathbf{x}_t; y_{\text{neg}}, t) \\
=& \omega_1 \cdot [(\epsilon_\phi(\mathbf{x}_t; y, t) - \epsilon_\phi(\mathbf{x}_t; t))] - \omega_2 \cdot [(\epsilon_\phi(\mathbf{x}_t; y_{\text{neg}}, t) - \epsilon_\phi(\mathbf{x}_t; t))] \\
=& \omega_1 \cdot \epsilon_\phi(\mathbf{x}_t; y, t) + (\omega_2 - \omega_1) \cdot \epsilon_\phi(\mathbf{x}_t; t) - \omega_2 \cdot \epsilon_\phi(\mathbf{x}_t; y_{\text{neg}}, t)
\end{aligned}
\tag{10}
$$

We use different weights $\omega_1$ and $\omega_2$ for the positive and negative prompts, respectively, to mitigate the negative effects of the latter. By gradually decreasing $\omega_2$, we observe that our modified loss function enhances both the quality and the fidelity of the texture results, as well as the alignment with the target prompt (refer to Fig. 5). Note that this simple formulation is based on our new interpretation of negative prompts from the perspective of Classifier Score Distillation, and it is not part of the original SDS framework.

**CSD for Text-Guided Editing**    The CSD method also enables us to perform text-guided 3D editing. Suppose we have a text prompt $y_{\text{target}}$ that describes the desired object and another text prompt $y_{\text{edit}}$ that specifies an attribute that we want to change in the original object. We can modify the loss function in Eq. (10) by replacing $y$ and $y_{\text{neg}}$ with $y_{\text{target}}$ and $y_{\text{edit}}$, respectively:

$$
\begin{aligned}
\delta_x^{\text{edit}} =& \omega_1 \cdot \delta_x^{\text{cls}}(\mathbf{x}_t; y_{\text{target}}, t) - \omega_2 \cdot \delta_x^{\text{cls}}(\mathbf{x}_t; y_{\text{edit}}, t) \\
=& \omega_1 \cdot \epsilon_\phi(\mathbf{x}_t; y_{\text{target}}, t) + (\omega_2 - \omega_1) \cdot \epsilon_\phi(\mathbf{x}_t; t) - \omega_2 \cdot \epsilon_\phi(\mathbf{x}_t; y_{\text{edit}}, t).
\end{aligned}
\tag{11}
$$

As shown in Fig. 6, this method allows us to edit the rendered image according to the target description $y_{\text{target}}$ while modifying or removing the attribute in $y_{\text{edit}}$. Text-to-3D generation can be seen as a special case of this method, where we set $\omega_2 = 0$ and no specific attribute in the original scene is given to be edited. Our method shares a similar formulation as Delta Denoising Score (Hertz et al., 2023) for image editing, which we discuss in the Appendix. Furthermore, we can adjust the weights $\omega_1$ and $\omega_2$ to balance the alignment with the text prompt and the fidelity to the original scene.

**Discussions and Connection to Variational Score Distillation (VSD)**    With CSD, the new variational score distillation (VSD) (Wang et al., 2023b) can be interpreted from the perspective of using a negative classifier score. VSD enhances SDS by replacing the noise term $\epsilon$ with the output of another text-conditioned diffusion model, denoted as $\epsilon_{\phi^*}$, which is simultaneously trained using the rendered images from the current scene states. When optimizing the 3D scene, the gradient direction applied to the rendered image is:

$$
\begin{aligned}
\delta_x^{\text{vsd}} =& [\epsilon_\phi(\mathbf{x}_t; y, t) - \epsilon_{\phi^*}(\mathbf{x}_t; y, t)] + \omega \cdot [\epsilon_\phi(\mathbf{x}_t; y, t) - \epsilon_\phi(\mathbf{x}_t; t)] \\
=& (\omega + 1) \cdot [\epsilon_\phi(\mathbf{x}_t; y, t) - \epsilon_\phi(\mathbf{x}_t; t)] - [\epsilon_{\phi^*}(\mathbf{x}_t; y, t) - \epsilon_\phi(\mathbf{x}_t; t)].
\end{aligned}
\tag{12}
$$

Since $\epsilon_{\phi^*}$ is obtained by fine-tuning the conditional branch of a pre-trained diffusion model using LoRA (Hu et al., 2021), we can assume that $\epsilon_\phi(\mathbf{x}_t; t) \approx \epsilon_{\phi^*}(\mathbf{x}_t; t)$ and then the term $[\epsilon_{\phi^*}(\mathbf{x}_t; y, t) - \epsilon_\phi(\mathbf{x}_t; t)]$ corresponds to predicting $\nabla_{\mathbf{x}_t} \log p_{\phi^*}(y|\mathbf{x}_t)$, where $p_{\phi^*}$ is a shifted distribution that adapts to the images rendered from the current 3D object. Thus, Eq. (12) encourages the optimization direction to escape from $p_{\phi^*}$ which can be interpreted as an ability to adaptively learn negative classifier scores instead of relying on a predefined negative prompt. However, this also introduces training inefficiency and instabilities. In our experiments, we find that CSD combined with general negative prompts can achieve high-quality texture quality comparable to VSD.

## 5    EXPERIMENTS

We evaluate the efficacy of our proposed Classifier Score Distillation method across three tasks: text-guided 3D generation, text-guided texture synthesis, and text-guided 3D editing. We present qualitative and quantitative analysis for text-guided 3D generation in Sec. 5.2 and text-guided texture synthesis in Sec. 5.3. To further substantiate the superiority of our approach, we conduct user studies for these two tasks. To showcase the capabilities of our formulation in 3D editing, illustrative examples are provided in Sec. 5.4.

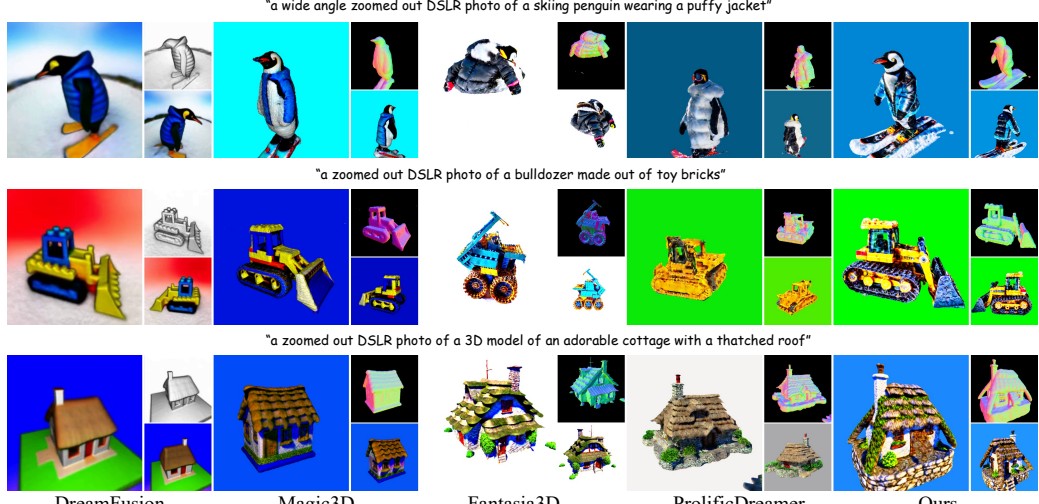

Figure 3: Qualitative comparisons to baselines for text-to-3D generation. Our method can generate 3D scenes that align well with input text prompts with realistic and detailed appearances.

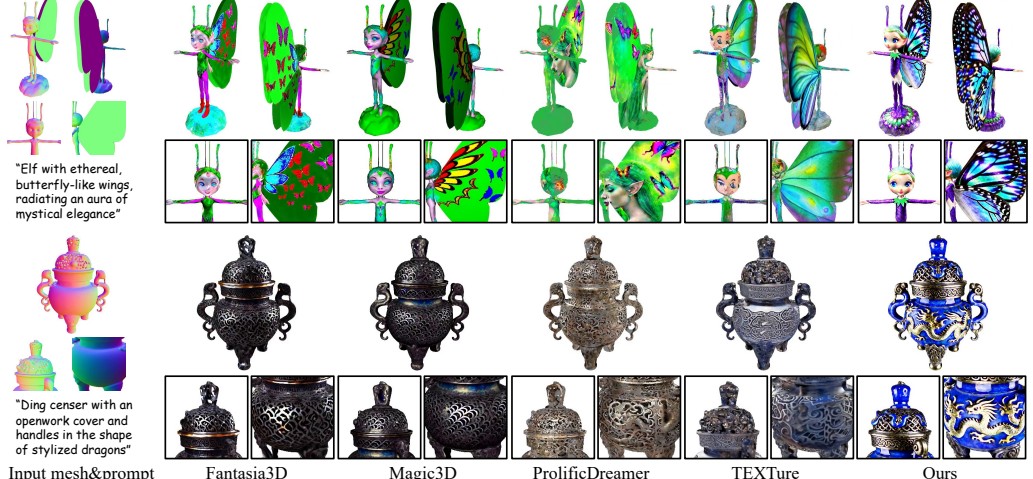

Figure 4: Qualitative comparisons to baselines for text-guided texture synthesis on 3D meshes. Our method generates more detailed and photo-realistic textures.

## 5.1 IMPLEMENTATION DETAILS

**Text-Guided 3D Generation**   We follow Magic3D (Lin et al., 2023) to initially generate a scene represented by Neural Radiance Fields (NeRF) using low-resolution renderings. Subsequently, the scene is converted into a triangular mesh via differentiable surface extraction (Shen et al., 2021) and further refined using high-resolution mesh renderings by differentiable rasterization (Laine et al., 2020). For the NeRF generation, we utilize the DeepFloyd-IF stage-I model (StabilityAI, 2023), and for the mesh refinement, we use the Stable Diffusion 2.1 model (Rombach et al., 2022) to enable high-resolution supervision. For both stages, CSD is used instead of SDS.

**Text-Guided Texture Synthesis**   Given a mesh geometry and a text prompt, we apply CSD to obtain a texture field represented by Instant-NGP (Müller et al., 2022). We employ ControlNets (Zhang & Agrawala, 2023) based on the Stable Diffusion 1.5 as our diffusion guidance since it can improve alignment between the generated textures and the underlying geometric structures. Specifically, we apply Canny edge ControlNet where the edge is extracted from the rendered normal maps, and depth ControlNet on rendered depth maps.

Table 1: User study on two tasks. In both tasks, more users prefer our results.

| Methods | Text-to-3D (%) ↑ | Texture Synthesis (%) ↑ |
|---|---|---|
| DreamFusion (Poole et al., 2022) | 30.3 | - |
| Magic3D (Lin et al., 2023) | 10.3 | 12.4 |
| Fantasia3D (Chen et al., 2023) | - | 11.0 |
| ProlificDreamer (Wang et al., 2023b) | - | 7.9 |
| TEXTure (Richardson et al., 2023) | - | 11.0 |
| Ours | **59.4** | **57.7** |

Table 2: Quantitative comparisons to baselines for text-to-3D generation, evaluated by CLIP Score and CLIP R-Precision.

| Methods | CLIP Score (↑) | CLIP R-Precision (%) ↑ | | |
|---|---|---|---|---|
| | | R@1 | R@5 | R@10 |
| DreamFusion (Poole et al., 2022) | 67.5 | 73.1 | 90.7 | **97.2** |
| Magic3D (Lin et al., 2023) | 74.9 | 74.1 | 91.7 | 96.6 |
| Ours | **78.6** | **81.8** | **94.8** | 96.3 |

## 5.2 TEXT-GUIDED 3D GENERATION

**Qualitative Comparisons.**  We present some representative results using CSD in Fig. 1, including both NeRF generation (a) and mesh refinement (b) results. In general, CSD can generate 3D scenes that align well with input text prompts with realistic and detailed appearances. Even only trained with low-resolution renderings (a), our results do not suffer from over-smoothness or over-saturation. We also find that our approach excels in grasping complex concepts in the input prompt, which is a missing property for many previous methods. In Fig. 3, we compare our generation results with DreamFusion (Poole et al., 2022), Magic3D (Lin et al., 2023), Fantasia3D (Chen et al., 2023), and ProlificDreamer (Wang et al., 2023b). For DreamFusion, we directly take the results provided on its official website. For the other methods, we obtain their results using the implementations from threestudio (Guo et al., 2023). Compared with SDS-based methods like DreamFusion, Magic3D, and Fantasia3D, our results have significantly more realistic appearances. Compared with ProlificDreamer which uses VSD, our approach can achieve competitive visual quality while being much faster, i.e., 1 hour on a single A800 GPU as opposed to 8 hours required by ProlificDreamer.

**Quantitative Evaluations.**  We follow previous work (Jain et al., 2022; Poole et al., 2022; Luo et al., 2023) to quantitatively evaluate the generation quality using CLIP Score (Hessel et al., 2022; Radford et al., 2021) and CLIP R-Precision (Park et al., 2021). Specifically, the CLIP Score measures the semantic similarity between the renderings of the generated 3D object and the input text prompt. CLIP R-Precision measures the top-N accuracy of retrieving the input text prompt from a set of candidate text prompts using CLIP Score. We generate 3D objects using 81 diverse text prompts from the website of DreamFusion. Each generated 3D object is rendered from 4 different views (front, back, left, right), and the CLIP Score is calculated by averaging the similarity between each rendered view and a text prompt. We use the CLIP ViT-B/32 (Radford et al., 2021) model to extract text and image features, and the results are shown in Tab. 2. Our approach significantly outperforms DreamFusion and Magic3D in terms of CLIP Score and CLIP R-Precision, indicating better alignment between the generated results and input text prompts.

**User Studies.**  We conduct a user study for a more comprehensive evaluation. We enlist 30 participants and ask them to compare multiple outputs, selecting the one they find most aligned with criteria such as visual quality and text alignment. In total, we collect 2289 responses. The results, presented in Tab. 1 (Text-to-3D), reveal that 59.4% of the responses prefer our results, demonstrating the superior quality of our approach.

## 5.3 TEXT-GUIDED TEXTURE SYNTHESIS

We select 20 diverse meshes from Objaverse (Deitke et al., 2022) for the texture generation task. We compare our generation results with those from Magic3D (Lin et al., 2023), Fantasia3D (Chen et al., 2023), ProlificDreamer (Wang et al., 2023b), and TEXTure (Richardson et al., 2023). We obtain the results of TEXTure using the official implementation and others using the implementations in threestudio. As illustrated in the Fig. 4, our method excels in generating photo-realistic texture details. Moreover, owing to our optimization based on implicit functions, the results we produce do not exhibit the seam artifacts commonly observed in TEXTure (Richardson et al., 2023). Our approach ensures both local and global consistency in the generated textures. We also conducted the user study, where 30 participants were asked and we got a total of 537 responses. The result is presented in Tab. 1 (Texture Synthesis), where 57.7% of the responses prefer our results.

Here, we examine how different uses of the negative prompt can impact visual quality. As shown in Fig. 5, incorporating negative prompts can indeed enhance the visual quality. However, we observe that it may compromise the alignment with the original text prompt, i.e., the original text prompt describes the color of the clothes as blue, but the generated clothes turn out to be white. As we have illustrated, since the negative prompt serves as an additional classifier score to guide the direction

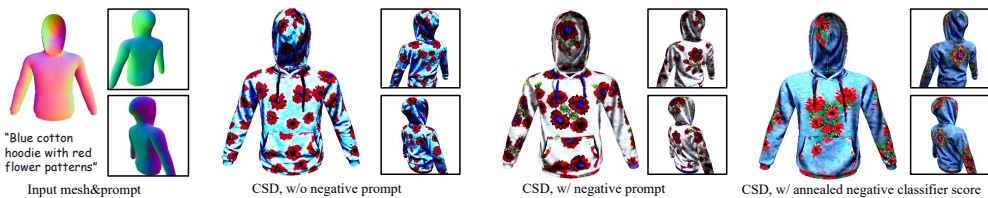

Figure 5: Ablation study on negative prompts and annealed negative classifier scores.

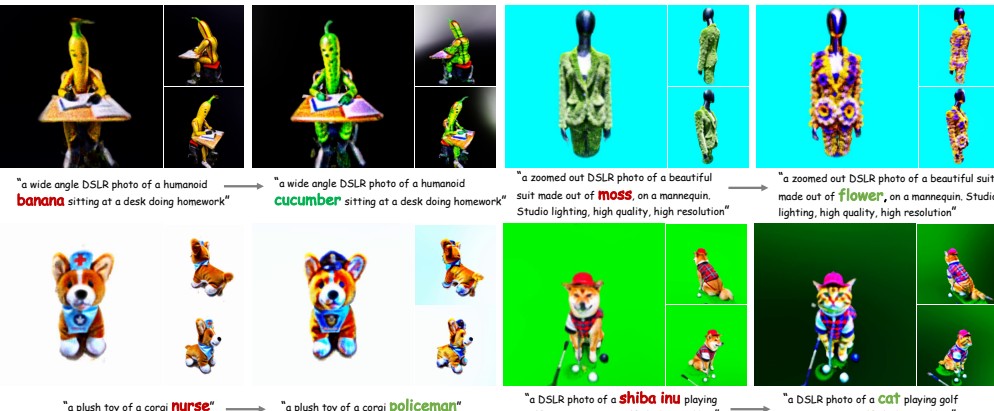

Figure 6: Demonstration of CSD in text-guided 3D editing Tasks. Our method effectively modifies attributes based on the given prompt while faithfully preserving the remaining features.

of the update, it has the effect of weakening the influence of the original prompt. Fortunately, our formulation allows for dynamic adjustment of weights to balance these forces. As shown in Fig. 5, we set $\omega_1 = 1$ and reduce the weight $\omega_2$ assigned to the negative classifier score, resulting in improved visual quality that remains aligned with the text prompt.

## 5.4 TEXT-GUIDED 3D EDITING

In Fig. 6, we demonstrate that CSD can be further applied to edit the existing NeRF scenes. For each scene, based on Eq. (11), we employ the portion that requires editing as the negative prompt (e.g., "nurse"), while modifying the original prompt to reflect the desired scene (e.g., "a plush toy of a corgi policeman"). We empirically set $\omega_1 = 1$ and $\omega_2 = 0.5$. Our approach yields satisfactory results, aligning edited objects with new prompts while maintaining other attributes.

## 6 CONCLUSION, DISCUSSION, AND LIMITATION

In this paper, we introduce Classifier Score Distillation (CSD), a novel framework for text-to-3D generation that achieves state-of-the-art results across multiple tasks. Our primary objective is to demonstrate that the classifier score, often undervalued in the practical implementations of SDS, may actually serve as the most essential component driving the optimization. The most significant implication of our results is to prompt a rethinking of what truly works in text-to-3D generation in practice. Building on the framework of CSD, we further establish connections with and provide new insights into existing techniques.

However, our work has certain limitations and opens up questions that constitute important directions for future study. First, while our empirical results show that using CSD leads to superior generation quality compared to SDS, we have not yet been able to formulate a distribution-based objective that guides this optimization. Second, despite achieving photo-realistic results in 3D tasks, we find that the application of CSD to 2D image optimization results in artifacts. We hypothesize that this discrepancy may arise from the use of implicit fields and multi-view optimization strategies in 3D tasks. Investigating the underlying reasons is of significant interest in the future. Despite these limitations, we believe our findings play a significant role in advancing the understanding of text-to-3D generation and offer a novel perspective for this field.

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

# A APPENDIX

## A.1 RELATED WORKS

**Text-Guided 3D Generation Using 2D Diffusion Models.** In recent years, after witnessing the success of text-to-image generation, text-to-3D content generation has also made stunning progress. Early work utilizes CLIP (Radford et al., 2021) to guide text-to-3D generation (Jain et al., 2022; Mohammad Khalid et al., 2022) or texture synthesize (Chen et al., 2022; Michel et al., 2022). Recently, utilizing 2D diffusion models pre-trained on massive text-image pairs (Schuhmann et al., 2022) for text-guided 3D generation has become mainstream. As the pioneering work of this field, DreamFusion (Poole et al., 2022) proposes the Score Distillation Sampling (SDS) technique (also known as Score Jacobian Chaining (SJC) (Wang et al., 2023a)), which is able to "distill" 3D information from 2D diffusion models. SDS has been widely used and discussed in the following works (Lin et al., 2023; Metzer et al., 2023; Chen et al., 2023; Wang et al., 2023b; Huang et al., 2023; Zhu & Zhuang, 2023; Shi et al., 2023), which attempt to improve DreamFusion in many different ways. Magic3D (Lin et al., 2023) and Fantasia3D (Chen et al., 2023) investigate the possibility of optimizing the mesh topology instead of NeRF for efficient optimization on high-resolution renderings. They operate on $512 \times 512$ renderings and achieve much more realistic appearance modeling than DreamFusion which is optimized on $64 \times 64$ NeRF renderings. MVDream Shi et al. (2023) alleviates the Janus problem of this line of methods by training and distilling a multi-view text-to-image diffusion model using renderings from synthetic 3D data. DreamTime (Huang et al., 2023) demonstrates the importance of diffusion timestep $t$ in score distillation and proposes a timestep annealing strategy to improve the generation quality. HiFA (Zhu & Zhuang, 2023) re-formulates the SDS loss and points out that it is equivalent to the MSE loss between the rendered image and the 1-step denoised image of the diffusion model. ProlificDreamer (Wang et al., 2023b) formulates the problem as sampling from a 3D distribution and proposes Variational Score Distillation (VSD). VSD treats the 3D scene as a random variable instead of a single point as in SDS, greatly improving the generation quality and diversity. However, VSD requires concurrently training another 2D diffusion model, making it suffer from long training time. 2D diffusion models can also be used for text-guided 3D editing tasks (Haque et al., 2023; Shao et al., 2023; Zhuang et al., 2023; Kim et al., 2023), and texture synthesis for 3D models (Richardson et al., 2023). In this paper, we provide a new perspective on understanding the SDS optimization process and derive a new optimization strategy named Classifier Score Distillation (CSD). CSD is a plug-and-play replacement for SDS, which achieves better text alignment and produces more realistic appearance in 3D generation. We also show the astonishing performance of CSD in texture synthesis and its potential to perform 3D editing.

## A.2 CSD FOR EDITING

Our method enables text-guided 3D editing. We observe that our formulation bears some resemblance to the Delta Denoising Score (DDS) (Hertz et al., 2023). Here, we demonstrate that these are two fundamentally different formulations and discuss their relations.

Specifically, DDS is proposed for 2D image editing. Given an initial image $\hat{\mathbf{x}}$, an initial prompt $\hat{y}$, and a target prompt $y$, assume that the target image to be optimized is $\mathbf{x}$. It utilizes SDS to estimate two scores. The first score is calculated using $\nabla_\theta \mathcal{L}_{\text{SDS}}(\mathbf{x}_t, y, t)$, while the second is determined by $\nabla_\theta \mathcal{L}_{\text{SDS}}(\hat{\mathbf{x}}_t, \hat{y}, t)$. They observed that relying solely on the first score would lead to blurred outputs and a loss of detail. Therefore, they decompose it into two implicit components: one is a desired direction guiding the image toward the closest match with the text, and another is an undesired component that interferes with the optimization process, causing some parts of the image to become smooth and blurry (Hertz et al., 2023). Their key insight is the belief that the score from $\nabla_\theta \mathcal{L}_{\text{SDS}}(\hat{\mathbf{x}}_t, \hat{y}, t)$ can describe the undesired component. Thus, they perform editing by subtracting this score during optimization. Using our notions, their editing objective is expressed as:

$$\delta_x^{\text{dds}} = \delta_x^{\text{sds}}(\mathbf{x}_t; y, t) - \delta_x^{\text{sds}}(\hat{\mathbf{x}}_t; \hat{y}, t) \tag{13}$$

Since the classifier-free guidance is still used here for both two SDS terms, we can look closer and formulate it as:

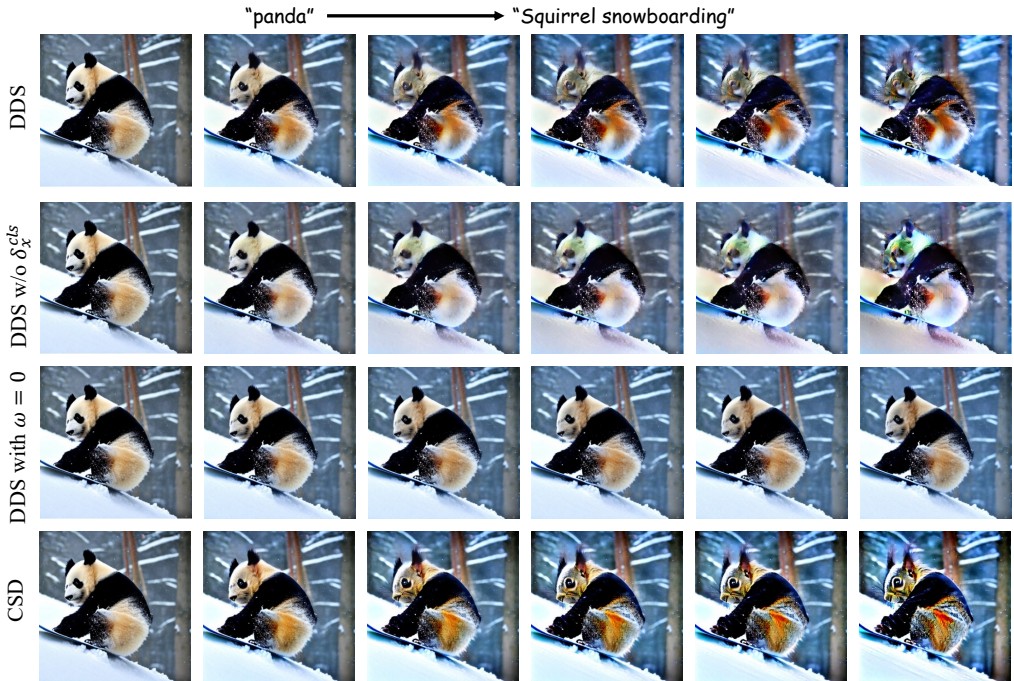

Figure 7: Comparisons between DDS variants and CSD on 2D image editing tasks, where the panda is edited to be a squirrel.

$$\delta_x^{\text{dds}} = [\epsilon_\phi(\mathbf{x}_t; y, t) - \epsilon_\phi(\hat{\mathbf{x}}_t; \hat{y}, t)] + \omega \cdot \underbrace{[\epsilon_\phi(\mathbf{x}_t; y, t) - \epsilon_\phi(\mathbf{x}_t; t)]}_{\delta_x^{\text{cls}}} - \omega \cdot [\epsilon_\phi(\hat{\mathbf{x}}_t; \hat{y}, t) - \epsilon_\phi(\hat{\mathbf{x}}_t; t)] \quad (14)$$

Note that the above gradient update rule actually consists of our classifier score $\delta_x^{\text{cls}}$, so it is interesting to see if the essential part to drive the optimization is still this component. To demonstrate this, we do several modifications to ablate the effect of $\delta_x^{\text{cls}}$ on 2D image editing. As shown in Fig. 7, without a guidance weight, DDS cannot do effective optimization. Besides, we ablate the effect of only removing $\delta_x^{\text{cls}}$ and find the optimization yet cannot be successful. Instead, solely relying on $\delta_x^{\text{cls}}$ can achieve superior results. Thus, we hypothesize the essential part of driving the optimization is $\delta_x^{\text{cls}}$. The insight here is also more straightforward than that of DDS and we do not need the reference image to provide a score that is considered an undesired component.

## A.3 ADDITIONAL RESULTS

**We provide an extensive gallery of video results in the supplementary material**. The gallary features a comprehensive comparison between our method and existing approaches. Specifically, the supplementary files include 81 results for the text-guided 3D generation task and 20 results for the text-guided texture synthesis task, along with comparisons with baseline methods. Please refer to geometry.html for comparisons on 3D generation and texture.html for comparisons on texture synthesis.

## A.4 IMPLEMENTATION DETAILS

For the generation of NeRF in the first stage, we utilize sparsity loss and orientation loss (Verbin et al., 2022) to constrain the geometry. The weight of the orientation loss linearly increases from 1 to 100 in the first 5000 steps, while the sparsity loss linearly decreases from 10 to 1 during the same period. After 5,000 iterations, we replace the RGB outputs as normal images with a probability of 0.5 for surface refinement. The weight of the negative classifier score is gradually annealed from

0 to 1 as the shape progressively takes form. To mitigate the Janus problem, we employ Perp-Neg (Armandpour et al., 2023) with a loss weight set to 3 and additionally constrain the camera view to focus only on the front and back views in the first 1,000 iterations. For prompts without clear directional objects, we omit Perp-Neg and use a larger batch size of 4. This extends training time by approximately 40 minutes. For the second stage of mesh refinement, we use a normal consistency loss with a weight of 10,000 and replace the RGB outputs as normal images with a probability of 0.5 for surface refinement. The weight of the negative classifier score is annealed from 1 to 0.5. For both stages, we use negative prompts "oversaturated color, ugly, tiling, low quality, noisy". For texture generation, we optimize over 30,000 iterations. To achieve better alignment of geometry and texture, we use ControlNets as guidance. Specifically, we employ canny-control and depth-control, where canny-control applies canny edge detection to the rendered normal image and uses the edge map as a condition. For both conditions, we start with a weight of 0.5 for the first 1000 steps, then reduce it to 0.2.

