# OpenReview forum: "Text-to-3D with Classifier Score Distillation"
_ICLR.cc/2024/Conference — ICLR 2024 poster_

### Official Review · Reviewer_FT6d · 2023-10-30

**Soundness:** 2 fair
**Presentation:** 2 fair
**Contribution:** 2 fair
**Rating:** 5
**Confidence:** 4

**Summary:**

This paper addresses the problem of text-to-3D, in which a 3D model of an object (represented by a NeRF) is produced given a text prompt describing the object. It shows that the classifier-free guidance part of the SDS loss is the main term driving the optimization of the NeRF, hence, proposing a new loss called Classifier Score Distillation (CSD). Furthermore, they also leverage the negative prompts to drive the rendered image away from low-quality region. In the experiment, the authors qualitatively show that the new CSD loss is easy to optimize as the SDS loss but bring the 3D model quality similar to the VSD (proposed in the Prolific Dreamer paper).

**Strengths:**

1. The paper is easy to understand and well-written.
2. The qualitative results are promising.
3. The proposed loss is simple and easy to reimplement.

**Weaknesses:**

1. The main weakness of this paper is that its reproducibility. Since the method is simple enough that I can reimplement it in the code base of SDS loss in the threestudio framework. However, I try my best to replicate every provided detail of the results are not good as shown in the paper. They are more or less like SDS, not good as VSD loss as claimed. Therefore, it would be much better if the authors do not provide their implementation to verify during the rebuttal phase, otherwise, it greatly affects their contribution.
2. In our reimplementation, the Janus problem is very serious.
3. Lack of quantitative comparison with SOTA approaches such as Prolific Dreamer, Fantasia3D....

**Questions:**

1. What negative prompts did you use?
2. How well does the CSD loss perform without the help of negative prompts. i.e., with the Eq. (7) only?
3. Which 81 text prompts you chose to compute CLIP R-precision, why don’t you compute all the text prompts (415 text prompts) provided in the DreamFusion repo?

---

> ### Author Response · Authors · 2023-11-22
> **Response to Reviewer FT6d (Part 1)**
>
> Dear Reviewer FT6d,
>
> We sincerely appreciate your great interest in our work and your thoughtful comments. In the response below, we address each of your concerns with the aim of clarifying and resolving any issues raised.
>
> ------
>
> **[W1] Implementation**
>
> Due to the change in the main loss function (SDS vs CSD), which obviates the need for a 'guidance scale', adjusting other parameters alongside the optimization process is necessary. We sincerely apologize for the lack of detailed hyperparameters. To address this, we have now revised and expanded the appendix to include these crucial details (i.e. appendix A.4). Additionally, we have made the code available at this **[anonymous link](https://drive.google.com/drive/folders/1_RsSkCnwA_y9Q0PYYlEBB5hIoIEA3OdC?usp=sharing)**, with the repository's README containing execution instructions. We will also release our code publicly.
>
> ------
>
> **[W2] Janus problem**
>
> 1. We would like to clarify that the core concepts and theoretical insights of our paper are not directly associated with the Janus problem. The fundamental challenge of the Janus problem stems not from the loss function, but from the limited 3D-aware capabilities inherent in the diffusion model. Our approach can be seamlessly integrated with techniques specifically designed to address the Janus problem, such as using the multi-view-diffusion model.
> 2. Here, we conduct a set of experiments using the pre-trained multi-view diffusion model from the concurrent work MVDream [1], combined with our CSD loss. We provide the qualitative results at this **[anonymous link](https://drive.google.com/drive/folders/1duCZB2KFozMGFOgQmdQGGiCwiZooEsOr?usp=sharing)**. The results illustrate that our loss function can generate realistic results, free from the Janus problem, indicating that this issue is more attributable to the diffusion model's 3D-awareness rather than our loss. **This set of experiments further strengthens the versatility and generalizability of our analysis.** We also provide the code at the same link in **W1** for your reference and we hope it can help the clarification.
>
> Reference:
>
> [1] MVDream: Multi-view Diffusion for 3D Generation. Shi et al., 2023

---

> ### Author Response · Authors · 2023-11-22
> **Response to Reviewer FT6d (Part 2)**
>
> **[W3] Quantitative comparisons with ProlificDreamer and Fantasia3D**
>
> 1. We would like to clarify that the reason that we omitted quantitative comparisons with ProlificDreamer and Fantasia 3D is that it is very time-consuming to evaluate ProlificDreamer based on public implementations (8h per prompt on an A800 GPU), and Fantasia3D requires manual tuning configuration parameters for each prompt (see their [note ("What do you want?")](https://github.com/Gorilla-Lab-SCUT/Fantasia3D#what-do-you-want) in their official implementation).  It is also worth noting that in their original paper, ProlificDreamer evaluated the performance on 15 prompts with a user study, and Fantasia3D evaluated the performance on 10 prompts with a user study.
>
> 2. Here, within the short period of rebuttal time and our constrained computational resources,  we randomly select 20 prompts and conduct experiments and comparisons with CSD, ProlificDreamer (using ThreeStudio), and Fantasia3D (using official code). The table below presents the results, showing that our method is generally comparable to Fantasia3D and ProlificDreamer. Moreover, our approach tends to perform better in comparison with other approaches under a more strict evaluation criterion like the CLIP score that directly measures the exact match between text prompts and generated images.  However, it's important to note that metrics like the CLIP score and R-precision primarily measure textual-image congruence and may not reflect 3D and texture quality. For a more comprehensive analysis, please refer to the detailed discussion below.
>
>    |                |         |                |                     |
>    | -------------- | :-----: | :------------: | :-----------------: |
>    |                | **CSD** | **Fantasia3D** | **ProlificDreamer** |
>    | **CLIP Score** |  81.6   |      77.8      |        80.2         |
>    | **CLIP R@1**   |  96.2   |      93.8      |        96.2         |
>    | **CLIP R@5**   |  98.8   |      100       |         100         |
>    | **CLIP R@10**  |   100   |      100       |         100         |
>    |                |         |                |                     |
>
> 3. **[Analysis of CLIP score for 3D evaluation]** It's important to note that in evaluating 3D synthesized results, the CLIP score may not fully reflect the quality for several reasons: (1) As an image-level metric, the CLIP score assesses each projected image independently, thus lacking comprehensiveness for 3D results; (2) **The CLIP score might not align with human perceptual judgments, which are crucial in evaluating generative results**; (3) This score evaluates the match between synthesis and text prompts using its learned feature space, which might not correlate directly with quality;  and (4) The CLIP score has biases toward its training data distribution, potentially differing from 3D projected images.  Consequently, many methods, like [1,2,3], choose to only conduct user studies for performance evaluation. In the final version, we will include user study results for these new experiments (note that we have previously conducted a user study comparing our results with DreamFusion and Magic3D, as detailed in our paper Table 1). Please see our qualitative results at the provided  [anonymous link](https://drive.google.com/drive/folders/1T6TuRrEH54CS4f84zcmf9sbOBeQ4OSMb?usp=sharing).
>
>
> Reference:
>
> [1] MVDream: Multi-view Diffusion for 3D Generation. Shi et al., 2023
>
> [2] ProlificDreamer: High-Fidelity and Diverse Text-to-3D Generation with Variational Score Distillation. Wang et al., 2023
>
> [3] Fantasia3D: Disentangling Geometry and Appearance for High-quality Text-to-3D Content Creation. Chen et al. 2023

---

> ### Author Response · Authors · 2023-11-22
> **Response to Reviewer FT6d (Part 3)**
>
> **[Q1, Q2] Negative prompts**
>
> 1. We use "oversaturated color, ugly, tiling, low quality, noisy" as our negative prompts. If negative prompts are not used, some generated results may show oversaturated colors or a higher occurrence of floaters. For demonstration, videos are provided at the following **[anonymous link](https://drive.google.com/drive/folders/1gvRq-Tr7dtTSBX-D3YRk184TEdIW1b_j?usp=sharing)**.
> 2. It should be emphasized that our paper's focus lies in providing a new understanding of score distillation and establishing the link between classifier scores and negative prompts rather than contributing to the technique of negative prompts itself. With this new understanding and guidance, we envision that future research could explore a range of techniques to improve quality, not limited to negative prompts. This includes developing adaptive negative classifier scores, for example, we examine the connection between VSD and classifier scores in equation (12).
>
> ------
>
> **[Q3] Using all 415 prompts for calculating CLIP R-precision**
>
> Thank you for your suggestion. For calculating the R-precision, we utilized the same set of prompts as those used in the generation process, in accordance with Cap3D [4]. These 81 prompts are detailed in our supplementary materials, with corresponding video filenames indicating the specific prompts employed. Note that our comparison between CSD and other methods is fair, since they are all based on same prompts.
>
> To further address your concern, we additionally calculated the R-precision using a larger set of 415 prompts for the caption set. As shown in the table below, the results are consistent with our original paper.
>
> |                |         |                 |             |
> | -------------- | :-----: | :-------------: | :---------: |
> |                | **CSD** | **DreamFusion** | **Magic3D** |
> | **CLIP Score** |  78.6   |      67.5       |    74.9     |
> | **CLIP R@1**   |  64.5   |      55.6       |    53.1     |
> | **CLIP R@5**   |  84.9   |      77.5       |    78.7     |
> | **CLIP R@10**  |  89.2   |      84.3       |    85.8     |
> |                |         |                 |             |
>
> Reference:
>
> [4] Scalable 3D Captioning with Pretrained Models. Luo et al., 2023
>
> ------
>
> We sincerely hope that our responses have adequately addressed your concerns and thank you again for your valuable suggestions that help us improve our work.

---

> ### Comment · Reviewer_FT6d · 2023-11-23
>
> I would like to thank the authors for their response. After running the provided code and viewing other supplemented videos from other answers, I have the following observations:
> 1. Regarding the 3d model quality if only optimized with 1 stage (NeRF only), the quality of CSD and SDS are similar. This is a fair comparison since the proposed method focused on the proposed loss, not on three-stage optimization. Similarly, we can take the results of SDS and then optimize further in 2 later stages (with exactly the same setting). Compared to VSD, the quality of CSD is fairly behind (also with 1-stage optimization).
> 2. Regarding the optimization time, it has a similar optimization time as SDS and is faster than VSD.
>
> Therefore, I can observably draw a conclusion that the proposed CSD is more or less similar to SDS, not a big improvement. Thus, I retain my initial rating.

---

> > ### Author Response · Authors · 2023-11-23
> >
> > Dear reviewer FT6d,
> >
> > 1. **Please note that the core focus of our work is 'why does SDS work?'. All aspects of our methodology section revolve around this central argument.** **As acknowledged by reviewer 8VxR, our paper is clearly motivated.** We have re-examined the **significant gap** between the theoretical derivation of probability density distillation (Equation (4)) and its practical implementation, which is a fundamental part of our study.
> > 2. The objective of our experimental section is to validate the discussions presented in our first point. **The focus is on demonstrating that 'using CFG alone can indeed work and yield good results,' rather than aiming to 'beat previous methods.'** Your observation that CSD and SDS appear similar actually reinforces our argument.
> > 3. Regarding VSD, as stated in our first point, the emphasis of our paper is on discussing why VSD works, rather than on developing a new method to outperform it. Our study and discussion aim to enrich the theoretical aspect of our work. We start from the concept of a negative classifier score and demonstrate that **VSD can be interpreted as CSD** with an adaptive negative classifier score. **Please note that this interpretation is entirely novel.**
> > 4. We strongly believe that understanding why the current range of score-distillation methods work is crucial. As acknowledged by reviewers pueN, 8VxR, and oCeh, they appreciated our new insights into score distillation (8VxR), the depth of our discussions (pueN), and the connections we established with other methods (oCeh). **Given that many studies focus primarily on modifying representations or training better diffusion models with more data, our work distinguishes itself as novel and unique.** We provide insights that operate on a different level, contributing a fresh perspective to the field.
> > 5. Additionally, regarding your mention of a 'three stage' and '2 later stages' in your comment, please note that our method involves only two stages. Concerning your comment about our method being faster than VSD, it's not just faster, but eight times faster. Furthermore, we don't believe that comparing only the first stage is fair, as VSD uses high-resolution diffusion, whereas we use low-resolution diffusion. Since we employ CSD throughout, a fair comparison should consider all stages of each method. It's also worth noting that VSD involves three stages, while our method comprises only two.
> > 6. **We sincerely hope that the main focus of our paper, which aims to present key insights and arguments, resonates with your considerations. Our intent is to contribute to a deeper understanding in the field, rather than just developing a method that surpasses previous ones.**

---

### Official Review · Reviewer_pueN · 2023-10-31

**Soundness:** 2 fair
**Presentation:** 3 good
**Contribution:** 3 good
**Rating:** 6
**Confidence:** 5

**Summary:**

This paper presents a text-to-3D generation model by exploring classifier-free guidance in score distillation. Experiments are conducted on several text-to-3D tasks to evaluate the proposal.

**Strengths:**

++ The main idea is simple yet effective for text-to-3D generation.

++ It is good to include an in-depth discussion about SDS in section 3.

++ Lots of promising qualitative results are shown to validate the effectiveness of proposal.

**Weaknesses:**

-- According to implementation details in section 5.1, this work uses two different pre-trained text-to-image models (DeepFloyd-IF stage-I model and Stable Diffusion 2.1). So is there any reason or ablation study for this design choice?

In addition, some baselines (like ProlificDreamer) only use the pre-trained text-to-image model of Stable Diffusion. It is somewhat no fair to compare this work with other baselines using different pre-trained models.

-- The evaluation of text-guided 3D generation is performed over 81 diverse text prompts from the website of DreamFusion. However, I noticed that the website of DreamFusion (https://dreamfusion3d.github.io/gallery.html) contains lots of results (more than 81 prompts). So how to choose the 81 diverse text prompts? Any screening criteria behind?

Moreover, this evaluation only uses CLIP ViT-B/32 to extract text and image features, while DreamFusion uses three models (CLIP B/32, CLIP B/16, CLIP L/14) to measure CLIP R-Precision. So following DreamFusion, it is better to report more results using more CLIP models.

-- The experimental results are somewhat not convincing, since the comparison of quantitative results is inadequate and more detailed experiments/baselines should be included:

1) For text-guided 3D generation, Table 2 only includes two baselines, while other strong baselines (Fantasia3D and ProlificDreamer) are missing.

2) Section 5.2 only mentions the computational cost of ProlificDreamer and this work. It is better to list the computational cost of each run.

3) For text-guided texture synthesis, a strong baseline [A] is missing for performance comparison. Moreover, only user study is performed for this task, and I am curious to see more quantitative comparison using the CLIP score or CLIP R-Precision.
[A] Lei J, Zhang Y, Jia K. Tango: Text-driven photorealistic and robust 3d stylization via lighting decomposition[J]. Advances in Neural Information Processing Systems, 2022, 35: 30923-30936.

-- I am curious to see more results by plugging the proposed CSD into more baselines (like  DreamFusion and ProlificDreamer).

**Questions:**

Please check the details in Weaknesses section, e.g., more clarification about implementation details and more experimental results.

---

> ### Author Response · Authors · 2023-11-22
> **Response to Reviewer pueN (Part 1)**
>
> Dear Reviewer pueN,
>
> We sincerely appreciate your insightful comments and valuable suggestions. In the response below, we address each of your concerns with the aim of clarifying and resolving any issues raised.
>
> ------
>
> **[W1] Reason for DeepFloyd-IF stage-I & Fairness of comparison**
>
> 1. The Stable Diffusion model is a high-resolution generative model (512x512), while the DeepFloyd-IF stage-I is a low-resolution model (64x64). Since our first stage focuses on generating rough results, we use DeepFloyd-IF stage-I.
> 2. We did our best to ensure fair comparisons with other methods by faithfully re-implementing their approaches based on their design principles or directly using their best available results as follows: 1) DreamFusion uses the superior **Imagen** model, which is not open-sourced. Fortunately, it publishes all results and thus, we directly compared our results with the best results from its official website. 2) The original Magic3D paper used **eDiff-I** in its first stage, which is also not open-sourced. Besides, Magic3D has not released its code or more results, so we re-implement its method upon the ThreeStudio. As the first stage of Magic3D also focuses on generating a coarse shape, we used DeepFloyd-IF stage-I in its first stage, the same as our setting. 3) ProlificDreamer needs high-resolution NeRF training in its first stage,  which requires high-resolution diffusion models. Thus, we followed their setting by using Stable Diffusion in the first stage. 4) Fantasia3D disentangles the generation into geometry and texture, in which both stages need high-resolution diffusion models to guarantee high-quality results. Thus, we followed their original paper by using Stable Diffusion.
>
> ------
>
> **[W2-1] Criteria of text prompts**
>
> 1. Thank you for this great question. Our criteria for selecting prompts followed three principles:
>    - **Sufficient Coverage and Diversity**: For example, some categories in the prompt set include: 1). **Animal Characters**: e.g., "a koala wearing a party hat"; 2). **Daily Life Objects**: e.g., "a shiny red stand mixer"; 3). **Transportation and Structures**: e.g., "a Panther De Ville car";  4). **Fantastical and Whimsical Themes**: e.g., "a humanoid banana doing homework"; and 5). **Nature and Scenery**: e.g., "a bumblebee sitting on a pink flower".
>    - **Avoidance of Overly Similar Prompts**: There are many very similar prompts on the official DreamFusion website. We filtered these out to avoid such situations and achieve greater diversity.
>    - **Inclusion of Uniquely Creative Prompts**: We retained prompts that are highly imaginative and creative, such as "a humanoid banana doing homework".
> 2. It's noteworthy that there is no established gold standard for evaluation prompts, and existing methods often select a subset of prompts for easier evaluation. For example, ProlificDreamer assessed their method using 15 prompts in a user study, while Fantasia3D did so with 10 prompts. Therefore, in our original paper, we adhered to this practice, choosing 81 prompts to guarantee coverage, diversity, and creativity, and to avoid redundancy in our representative evaluation prompts.
> 3. Given the constraints in computational resources, the brief rebuttal period, and the demands of conducting other experiments, generating results for 415 prompts exceeds our current capacity within the rebuttal timeframe. However, we have included results from additional randomly selected prompts, as illustrated in the comparison in **W3-1**. We pledge to release our results for all prompts once the experiments are completed.
>
> ------
>
> **[W2-2]  More CLIP R-precision**
>
> We report more CLIP scores and CLIP R-precision here using different pre-trained models. Please refer to **W3-1** for an analysis of the CLIP score.
>
> |                           |         |                 |             |
> | ------------------------- | :-----: | :-------------: | :---------: |
> |                           | **CSD** | **DreamFusion** | **Magic3D** |
> | **CLIP Score (ViT-L/14)** |  69.1   |      63.6       |    65.4     |
> | CLIP R@1 (ViT-L/14)       |  96.0   |      88.3       |    84.9     |
> | CLIP R@5 (ViT-L/14)       |  99.1   |      97.8       |    97.5     |
> | CLIP R@10 (ViT-L/14)      |  99.7   |      98.8       |    99.4     |
> | **CLIP Score (ViT-B/32)** |  78.6   |      67.5       |    74.9     |
> | CLIP R@1 (ViT-B/32)       |  81.8   |      73.1       |    74.1     |
> | CLIP R@5 (ViT-B/32)       |  94.8   |      90.7       |    91.7     |
> | CLIP R@10 (ViT-B/32)      |  96.3   |      97.2       |    96.6     |
> | **CLIP Score (ViT-B/16)** |  81.1   |      73.0       |    77.8     |
> | CLIP R@1 (ViT-B/16)       |  94.1   |      88.3       |    84.3     |
> | CLIP R@5 (ViT-B/16)       |  98.5   |      97.8       |    96.0     |
> | CLIP R@10 (ViT-B/16)      |  98.8   |      99.1       |    97.5     |
> |                           |         |                 |             |
>
> ------

---

> ### Author Response · Authors · 2023-11-22
> **Response to Reviewer pueN (Part 2)**
>
> **[W3-1] Comparisons with ProlificDreamer and Fantasia3D:**
>
> 1. We would like to clarify that the reason that we omitted quantitative comparisons with ProlificDreamer and Fantasia 3D is that it is very time-consuming to evaluate ProlificDreamer based on public implementations (8h per prompt on an A800 GPU), and Fantasia3D requires manual tuning configuration parameters for each prompt (see their [note ("What do you want?")](https://github.com/Gorilla-Lab-SCUT/Fantasia3D#what-do-you-want) in their official implementation). It is also worth noting that in their original paper, ProlificDreamer evaluated the performance on 15 prompts with a user study, and Fantasia3D evaluated the performance on 10 prompts with a user study.
>
> 2. Here, within the short period of rebuttal time and our constrained computational resources, we randomly select 20 prompts and conduct experiments and comparisons with CSD, ProlificDreamer (using ThreeStudio), and Fantasia3D (using official code). The table below presents the results, showing that our method is generally comparable to Fantasia3D and ProlificDreamer. Moreover, our approach tends to perform better in comparison with other approaches under a more strict evaluation criterion like the CLIP score that directly measures the exact match between text prompts and generated images. However, it's important to note that metrics like the CLIP score and R-precision primarily measure textual-image congruence and may not reflect 3D and texture quality. For a more comprehensive analysis, please refer to the detailed discussion below.
>
>    |                           |         |                |                     |
>    | ------------------------- | :-----: | :------------: | :-----------------: |
>    |                           | **CSD** | **Fantasia3D** | **ProlificDreamer** |
>    | **CLIP Score (ViT-L/14)** |  71.8   |      65.2      |        72.7         |
>    | CLIP R@1 (ViT-L/14)       |   100   |      93.8      |         100         |
>    | CLIP R@5 (ViT-L/14)       |   100   |      100       |         100         |
>    | CLIP R@10 (ViT-L/14)      |   100   |      100       |         100         |
>    | **CLIP Score (ViT-B/32)** |  81.6   |      77.8      |        80.2         |
>    | CLIP R@1 (ViT-B/32)       |  96.2   |      93.8      |        96.2         |
>    | CLIP R@5 (ViT-B/32)       |  98.8   |      100       |         100         |
>    | CLIP R@10 (ViT-B/32)      |   100   |      100       |         100         |
>    | **CLIP Score (ViT-B/16)** |  83.1   |      78.6      |        83.2         |
>    | CLIP R@1 (ViT-B/16)       |  97.5   |      96.2      |         100         |
>    | CLIP R@5 (ViT-B/16)       |   100   |      100       |         100         |
>    | CLIP R@10 (ViT-B/16)      |   100   |      100       |         100         |
>    |                           |         |                |                     |
>
> 3. **[Analysis of CLIP score for 3D evaluation]** It's important to note that in evaluating 3D synthesized results, the CLIP score may not fully reflect the quality for several reasons: (1) As an image-level metric, the CLIP score assesses each projected image independently, thus lacking comprehensiveness for 3D results; (2) **The CLIP score might not align with human perceptual judgments, which are crucial in evaluating generative results**; (3) This score evaluates the match between synthesis and text prompts using its learned feature space, which might not correlate directly with quality;  and (4) The CLIP score has biases toward its training data distribution, potentially differing from 3D projected images.  Consequently, many methods, like [1,2,3], choose to only conduct user studies for performance evaluation. In the final version, we will include user study results for these new experiments (note that we have previously conducted a user study comparing our results with DreamFusion and Magic3D, as detailed in our paper Table 1). Please see our qualitative results at the provided  [anonymous link](https://drive.google.com/drive/folders/1T6TuRrEH54CS4f84zcmf9sbOBeQ4OSMb?usp=sharing).
>
> Reference:
>
> [1] MVDream: Multi-view Diffusion for 3D Generation. Shi et al., 2023
>
> [2] ProlificDreamer: High-Fidelity and Diverse Text-to-3D Generation with Variational Score Distillation. Wang et al., 2023
>
> [3] Fantasia3D: Disentangling Geometry and Appearance for High-quality Text-to-3D Content Creation. Chen et al. 2023
>
> ------

---

> ### Author Response · Authors · 2023-11-22
> **Response to Reviewer pueN (Part 3)**
>
> **[W3-2] Computational cost for each run:**
>
> 1). For DreamFusion, we used their official website's results, employing 4 TPUv4 chips for 15,000 iterations over 1.5 hours (6 TPUv4 hours). 2). For Magic3D, we used a single A800 GPU, with a batch size of 4 for 10,000 NeRF iterations (70 minutes) and a batch size of 1 for 10,000 iterations in the second stage (20 minutes), totaling 1.5 GPU hours. 3). For Fantasia3D, following their official code, we used 8 GPUs, each stage taking about 10 minutes, for a total of 2.7 GPU hours. 4). For ProlificDreamer, we use their original paper's settings: 25,000 VSD iterations for NeRF (3.5 hours), 15,000 SDS iterations for geometry (30 minutes), and 30,000 VSD iterations for texture (4 hours), totaling 8 GPU hours. 5). For our experiments, both stages were optimized for 10,000 iterations, taking 35 and 25 minutes respectively, amounting to 1 GPU hour.
>
> ------
>
> **[W3-3] Comparisons with Tango & quantitative evaluations:**
>
> Thank you for your reminder. We have cited this paper in our revised paper (appendix A.1) and here we present the comparison results with it. We show the qualitative comparison results at this **[anonymous link](https://drive.google.com/drive/folders/1dANkz6X56wFKb-1jVakCsFimo_SuBVnD?usp=sharing)**.
>
> Regarding the task of texture synthesis, as mentioned in **W3-1**, evaluations based on CLIP may not be accurate. Most current works rely on user studies. Here, we provide the CLIP score for your reference.  Although Fantasia3D performs the best on two backbones, there is not a single method that wins in all metrics, and our method performs reasonably well on all evaluated settings.  **More importantly, our user study in Table 1 in our main paper demonstrates a clear win of our approach in terms of texture synthesis when compared to Fantasia 3D, ProlificDreamer, Magic3D, and TEXTure.**
>
> |                     |                           |                           |                           |
> | ------------------- | :-----------------------: | :-----------------------: | :-----------------------: |
> |                     | **CLIP Score (ViT-L/14)** | **CLIP Score (ViT-B/32)** | **CLIP Score (ViT-B/16)** |
> | **CSD**             |           64.6            |           72.7            |           74.5            |
> | **Tango**           |           61.6            |           72.3            |           74.0            |
> | **TEXTure**         |           61.4            |           72.0            |           72.6            |
> | **ProlificDreamer** |           63.8            |           73.1            |           75.0            |
> | **Fantasia3D**      |           63.2            |           74.2            |           75.3            |
> | **Magic3D**         |           61.9            |           71.1            |           74.2            |
> |                     |                           |                           |                           |
>
> ------
>
> **[W4] CSD on more baselines:**
>
> 1. Please note that for DreamFusion, ProlificDreamer, and our work, the principal differences and unique contributions are centered on the three distinct losses: SDS, VSD, and CSD. Therefore, integrating CSD into these methods essentially amounts to employing the CSD approach, albeit with some variations in the training process. For more details, kindly refer to equations 6, 8, and 12 in our paper. In fact, our loss is model-agnostic or independent of 3D representation, as evidenced by our approach in different stages. For instance, in the first stage, we utilize the DeepFloyd-IF stage-I model with NeRF representation, and in the second stage, we apply Stable Diffusion with DMTet representation. This showcases the versatility of our loss across various models and representations.
> 2. To show more results in more baselines, we integrated the proposed CSD into the pre-trained multi-view diffusion model from the recent concurrent work MVDream [1]. We provide the qualitative results at this [anonymous link](https://drive.google.com/drive/folders/1duCZB2KFozMGFOgQmdQGGiCwiZooEsOr?usp=sharing). The results indicate that our loss can be seamlessly integrated into this new framework, producing realistic contents and high-quality textures.
>
> Reference:
>
> [1] MVDream: Multi-view Diffusion for 3D Generation. Shi et al., 2023
>
> ------
>
> We sincerely hope that our answers have addressed your concerns and thank you again for your valuable suggestions that help us improve our work.

---

### Official Review · Reviewer_8VxR · 2023-10-31

**Soundness:** 3 good
**Presentation:** 4 excellent
**Contribution:** 3 good
**Rating:** 8
**Confidence:** 4

**Summary:**

This manuscript introduces a novel perspective on score distillation sampling (SDS). Classifier free guidance (CFG) can be interpreted as an implicit classifier based on the diffusion model that scores how much the image corresponds to the text. Empirically, SDS adds a CFG term to its gradient to ensure that the generation corresponds to the text prompt. However, by doing so, the gradients used in SDS in practice are dominated by this CFG term. This work proposes Classifier Score Distillation (CSD) which uses solely this CFG term to provide the gradients. This paper shows that CSD alone is sufficient to guide 3D generation. Furthermore, this work uses its CSD formulation to give a new interpretation of negative prompting with CFG and proposes a new negative prompting formulation that allows for explicit weights on both the positive and the negative directions.  This paper compares CSD both qualitatively and quantitatively to numerous baselines on multiple generation tasks showing SOTA performance. This work also shows CSD on editing tasks.

**Strengths:**

- Novel formulation of score distillation that gives an interesting new perspective.
- CSD is general and can be used for any approaches using score distillation (text-to-3D, text-driven image editing) to improve results. It can also be seamlessly integrated into any existing score distillation approaches.
- Thorough evaluation shows that CSD gives improvement over SDS both qualitatively and quantitatively.
- The paper is well written, clearly motivating and explaining the intuition behind CSD.

**Weaknesses:**

Major:
- This likely inherits the weaknesses of using a high CFG with standard SDS (I assume the following are true, but see questions for more details): less diversity of generations for a given prompt, less realistic generations, over saturated colors. [1]
- If I understand correctly, empirically, this is not much different than using SDS with a large weight for CFG. It would be helpful to show comparisons to SDS with very large CFG weights. See questions for more details.

Minor:
- Figure 2a: It might be more clear to show both norms on the same scale. At first glance it can be confusing if you don’t notice the different scales.
- Figure 2b: Consider including CSD here. It would be interested to see higher values for w as well since DreamFusion uses w=100.

References: [1] Wang, Zhengyi, Cheng Lu, Yikai Wang, Fan Bao, Chongxuan Li, Hang Su, and Jun Zhu. "ProlificDreamer: High-Fidelity and Diverse Text-to-3D Generation with Variational Score Distillation." arXiv preprint arXiv:2305.16213 (2023).

**Questions:**

- Does using CSD cause images to be less “realistic” since it removes the prior term of the loss? I.e. the generation will adhere to the text prompt very closely, but lead to an potentially unrealistic result?
- Similarly, how is the diversity of generations using CSD for a given prompt? I would guess that there is less diversity than SDS since higher CFG weight typically reduces diversity.
- What are the CFG weights used in the experiments section for the SDS on the baseline methods? It is specified that the DreamFusion results were obtained from its website implying a CFG of 100, but what about for the others? The default value in ThreeStudio appears to be 100 for methods using stable diffusion and 7.5 for Prolific Dreamer. Is that what was used for the experiments? If so, it might be helpful to add experiments showing existing SDS methods with very large CFG weights (i.e. 200, 500, 1000, etc.) and see how that compares to CSD.

**Details Of Ethics Concerns:**

No ethical concerns.

---

> ### Author Response · Authors · 2023-11-22
> **Response to Reviewer 8VxR (Part 1)**
>
> Dear Reviewer 8VxR,
>
> We sincerely appreciate your insightful comments and valuable suggestions. In the response below, we address each of your concerns with the aim of clarifying and resolving any issues raised.
>
> ------
>
> **[W1, Q1] Realistic**
>
> Thank you for your insightful comments. In our observations, results from CSD are **not** less realistic or more prone to over-saturation compared to the SDS. Indeed, negative prompts can generally be utilized to address these issues. Further, following the concept of negative classifier scores, we have presented a new insight in Equation (12) of our paper. We suggest that the reason VSD helps alleviate over-saturation might be due to its new gradient functioning like an adaptive negative score.
>
> ------
>
> **[W1, Q2] Diversity**
>
> 1. Empirically, we did **not** observe lower diversity using CSD compared to SDS. We think that improving diversity through reducing CFG  might not be a viable choice for the SDS-based method as a lower CFG value may also lead to optimization failure, which is verified by our analysis in Figure 2 of our paper. In this work, our focus is to uncover the component— the classifier-free guidance— that makes SDS work.
>
> 2. To improve diversity, we believe a better approach might be to consider more diverse prompts. We can enhance the user-provided prompts, ensuring they meet the user’s content requirements while introducing a variety of variations. This can be achieved automatically using large language models (LLMs). As an illustration, here we employed ChatGPT to enhance the given prompt 'a DSLR photo of a bear dressed in medieval armor', where we asked it to generate different variations while retaining the original meaning, resulting in the following prompts:
>
>    - “A DSLR photo of a panda bear in stylized Eastern medieval armor, reflecting Asian historical designs.”
>    - “A DSLR photo of a polar bear in ornate, silver-colored medieval armor, contrasting with its white fur.”
>    - “A DSLR photo of a sun bear wearing lightweight leather armor, reminiscent of medieval archers, with a quiver of arrows.”
>    - “A DSLR photo of a panda bear in ancient Chinese warrior armor, embellished with dragon motifs and vibrant colors.“
>
>    We then applied our method (NeRF stage) to these prompts, and we provided video results at the following **[anonymous link](https://drive.google.com/drive/folders/1q49ztkSIx9USFzeH1sFPSAVS54tDcCcR?usp=sharing)**. This approach successfully enhances diversity while not suffering from optimization issues.

---

> ### Author Response · Authors · 2023-11-22
> **Response to Reviewer 8VxR (Part 2)**
>
> **[W2, Q3] CFG in baseline methods & Experiment with larger CFG for SDS**
>
> 1. **For the CFG scale of other methods, we followed the values stated in their respective papers**. For ProlificDreamer, we set the CFG scale to 7.5 in both the first and third stages, where VSD is used, and increased it to 100 in the second stage, where SDS is employed for refining the geometry. For Fantasia3D, we consistently applied a CFG scale of 100 for both stages. In the case of Magic3D, we used a CFG scale of 100 for the second stage. However, for the first stage of Magic3D, since we do not have access to their unreleased eDiff-I model, we utilized the DeepFloyd-IF stage-I model with a CFG scale of 20, following the default implementation in ThreeStudio.
>
> 2. Although SDS with a large CFG value approaching infinity may approximate the effects of using classifier-free guidance alone (i.e., CSD), this raises significant optimization issues, making it impractical. Within the SDS framework, which combines generative and classifier priors, hyperparameter tuning with a large CFG scale becomes complex and often requires extensive trial and error to adjust learning rates and geometric constraint loss weights. For instance, an excessively large CFG scale can result in problems such as numerical overflow in gradient calculations due to substantial guidance weights. Moreover, this can cause a "gradient domination" effect, diminishing the impact of other geometric regularization terms.
> 3. To empirically demonstrate the effect of using higher CFG values with SDS loss, we have conducted experiments using the first stage of Magic3D. Note that the first stage of Magic3D is essentially the same as DreamFusion, with the difference being that DreamFusion employs Mip-NeRF representation and the more powerful Imagen model that we do not have access to. We use SDS with CFG scales of 20, 100, 200, 500, and 1000. However, simply increasing guidance weights does not guarantee satisfactory results, as confirmed in our results (refer to results in  **[anonymous link](https://drive.google.com/drive/folders/1qV2K2eaKibcpuOYkUEm3j7NdnK0wWzr5?usp=sharing)**): higher CFG values can lead to more floaters, and at extremely high CFG values, there's a risk of results collapsing.
>
> ------
>
> **[W3, W4] Revise of Figure 2**
>
> Thanks for your suggestion, we have revised the Figure 2 for better demonstration.
>
> ------
>
> We sincerely hope that our answers have addressed your concerns and thank you again for your valuable suggestions that help us improve our work.

---

### Official Review · Reviewer_oCeh · 2023-10-31

**Soundness:** 3 good
**Presentation:** 4 excellent
**Contribution:** 3 good
**Rating:** 8
**Confidence:** 4

**Summary:**

This paper introduces a new score distillation scheme for text-to-3D generation, dubbed, Classifier Score Distillation (CSD). While the original Score Distillation Sampling (SDS) from DreamFusion subtracts random noise, CSD subtracts unconditional noise estimate (or noise estimation with negative prompts). With CSD, the author shows its effectiveness in text-to-3D generation and texture synthesis.

**Strengths:**

- CSD is a simple yet effective method in transferring 2D diffusion prior to the 3D scene generation or editing. In contrast to prior state-of-the-art ProlificDreamer, it does not require fine-tuning of diffusion models, which may introduce training inefficiency and instabilities.

- The qualitative and quantitative results show its effectiveness compared to prior methods. Also, this work presents a relationship between Delta Denoising Score which also used subtraction of noises in image editing tasks. I believe this is also related to the noise subtraction scheme in collaborative score distillation [https://arxiv.org/abs/2307.04787] paper, which the discussion will make the paper more complete.

**Weaknesses:**

- In general, I do not see a crucial weakness of this paper as it illustrates a simple method that improves the current text-to-3D generation. I believe providing detailed hyperparameter ablation study will make  the paper more informative.

**Questions:**

- See Strengths; how the image-conditioned noise subtraction of InstructPix2Pix diffusion model in Collaborative Score Distillation paper can be related to classifier score distillation? Can Collaborative score distillation can be improved with classifier score distillation like approach?

---

> ### Author Response · Authors · 2023-11-22
> **Response to Reviewer oCeh**
>
> Dear Reviewer oCeh,
>
> We sincerely appreciate your insightful comments and valuable suggestions. In the response below, we address each of your concerns with the aim of clarifying and resolving any issues raised.
>
> ---
>
> **[S2, Q1] Relation to Collaborative Score Distillation**
>
> Thank you for your insightful comment and the recommended work, which we have now cited in our revised paper. We find it to be a very interesting work! Interestingly, its formulation also replaces subtracting noise with subtracting a score computed from a reference image. This bears a resemblance to our discussion on DDS. We believe that substituting this aspect with CSD could potentially further enhance the quality.
>
> Since the code for this paper has not been released yet, it is a bit challenging for us to faithfully reproduce this work during this short rebuttal period.  We will continue to pay close attention to this work and its release. Thank you for your insightful comments!
>
> To demonstrate the generality and adaptivity of our CSD formulation, we conducted additional experiments on the recently released multi-view diffusion model [1]. We provide the qualitative results at this **[anonymous link](https://drive.google.com/drive/folders/1duCZB2KFozMGFOgQmdQGGiCwiZooEsOr?usp=sharing)**. The results demonstrate that our loss can be seamlessly integrated into this new framework, producing realistic 3D content.
>
> Reference:
>
> [1] MVDream: Multi-view Diffusion for 3D Generation. Shi et al., 2023
>
> ---
>
> **[W1] Hyper-parameter ablation study**
>
> Thanks for your suggestion. In our experiment, we found that employing appropriate sparsity and orientation losses during the NeRF optimization is crucial for geometry. Here, we show four distinct experiments to demonstrate this: A). Without any geometric constraint loss. B). Constrained only by sparsity loss. C). Constrained only by orientation loss. D). (i.e., CSD) Employing both constraints simultaneously.
>
> We have provided video comparison results at this **[anonymous link](https://drive.google.com/drive/folders/1IU583pgt_2hipKA9W2Y7jnt8I_EzUubf?usp=drive_link)**. As can be observed, it's only when both constraints are applied together that we can ensure smoother normals and better 3D consistency (i.e., less Janus problem), as well as guarantee the overall success of the optimization process.
>
> ---
>
> We sincerely hope that our answers have addressed your concerns and thank you again for your valuable suggestions that help us improve our work.

---

### Author Response · Authors · 2023-11-22
**Response to AC and reviewers**

Dear reviewers and AC,

We sincerely appreciate your time and effort in reviewing our work and providing valuable suggestions.

As pointed out and commended by all the reviewers (FT6d, pueN, 8VxR, oCeh), our method is remarkably simple yet effective, and we have demonstrated this through a plethora of qualitative results. More importantly, the reviewers (pueN, 8VxR, oCeh) recognized and appreciated our new insights into score distillation (8VxR), the depth of our discussions (pueN), and the connections we established with other methods (oCeh). **This aspect forms the main focus of our paper and is a key part of what we believe contributes significantly to the field. We believe our work prompts a rethinking of what truly works in text-to-3D generation in practice, fostering the development of new theories and paradigms to guide the evolution of 3D generation in the future.**

The individual questions raised by the reviewers have been thoroughly addressed in our detailed responses to each reviewer.

---

### Meta-Review · Area_Chair_Juwd · 2023-12-12

**Metareview:**

The paper has received recommendations from three reviewers and one borderline rejection. The reviewers generally agree that the paper presents insightful discussions on why SDS (Score Distillation Sampling) works for 3D generation by explicitly writing out terms in classifier-free guidance. The authors also introduce the CSD (Classifier Score Distillation) formulation, where they observe the classifier score term to be critical for the success of practical SDS formulations with a large CFG (Classifier-Free Guidance) and propose the CSD as an alternative.

The primary concern raised by reviewer 8VxR and FT6dis whether the CSD formulation is significantly better than SDS and VSD. The reviewer, FT6d, implemented CSD and did not find it superior to VSD. The authors responded with additional analysis and videos to demonstrate the performance of CSD. However, concerns remain , with reviewer 8VxR arguing that the CSD formulation is similar to SDS with an infinite CFG scale.

After careful consideration of the reviews, rebuttal, and additional materials, the Area Chair (AC) concurs with the concerns raised by reviewers. The AC notes that the CSD formulation can still lead to over-saturation, akin to SDS with a high guidance scale. While the chosen negative prompt in this paper might mitigate this issue to some extent, the fundamental problem is not entirely resolved. The vanilla CSD, without a negative prompt, may inherit issues from SDS when the hyper-parameters of SDS are properly set.

Despite these concerns, the AC acknowledges the paper's interesting formula for analyzing SDS in text-to-3D generation. The AC also agrees with reviewers 8VxR and FT6d that the major weakness lies in the fact that the CSD formulation does not fully address the issues of diversity and over-saturation inherited from SDS. However, the AC does not find these issues significant enough to overturn the majority assessments, and therefore, recommends accepting the paper.

**Justification For Why Not Higher Score:**

The paper initiates an insightful discussion about why SDS works and has a reasonable analysis. However, the proposed method does not fully address the issues in SDS. Therefore, the AC recommends accepting the paper with poster.

**Justification For Why Not Lower Score:**

While the proposed method does not address the issues in SDS, the AC still finds the discussion in this paper worth sharing to the research community.

---

### Decision · Program_Chairs · 2024-01-16

Accept (poster)